# Multi-Prompt Alignment for Multi-Source Unsupervised Domain Adaptation

**Haoran Chen**[1,2]    **Xintong Han**[3]    **Zuxuan Wu**[1,2†]    **Yu-Gang Jiang**[1,2]

[1]Shanghai Key Lab of Intell. Info. Processing, School of CS, Fudan University
[2]Shanghai Collaborative Innovation Center of Intelligent Visual Computing
[3]Huya Inc

## Abstract

Most existing methods for unsupervised domain adaptation (UDA) rely on a shared network to extract domain-invariant features. However, when facing multiple source domains, optimizing such a network involves updating the parameters of the entire network, making it both computationally expensive and challenging, particularly when coupled with min-max objectives. Inspired by recent advances in prompt learning that adapts high-capacity models for downstream tasks in a computationally economic way, we introduce **M**ulti-**P**rompt **A**lignment (MPA), a simple yet efficient framework for multi-source UDA. Given a source and target domain pair, MPA first trains an individual prompt to minimize the domain gap through a contrastive loss. Then, MPA denoises the learned prompts through an auto-encoding process and aligns them by maximizing the agreement of all the reconstructed prompts. Moreover, we show that the resulting subspace acquired from the auto-encoding process can easily generalize to a streamlined set of target domains, making our method more efficient for practical usage. Extensive experiments show that MPA achieves state-of-the-art results on three popular datasets with an impressive average accuracy of 54.1% on DomainNet.

## 1   Introduction

Deep learning has achieved remarkable progress in various computer vision tasks [15, 11, 31, 24]. However, the success usually relies on supervised training using a massive amount of manually labeled data, which are often expensive and time-consuming to collect. Furthermore, current deep models are brittle to the presence of domain shift [27, 34, 47] in the forms of different image styles, varied lighting conditions, diverse viewpoints, *etc.*, between training and testing distributions.

Unsupervised domain adaptation (UDA) is a popular strategy that mitigates domain discrepancies by transferring knowledge learned from a well-labeled source domain to an unlabeled target domain [25, 4, 39, 42, 23]. While significant advances have been achieved, current approaches focus on the single source setting, where all the labeled training data share the same distribution. In practice, however, it is more common for the labeled data to be collected from multiple sources that are diverse in distribution. Naturally, one could still tackle this problem by straightforwardly combining all the data into one single source and applying off-the-shelf UDA methods. However, directly applying single-source UDA methods often results in limited performance, as domain shift also exists among different source domains.

The integration of multiple source domains for improved adaptation results on the unlabeled target domain is generally known as multi-source unsupervised domain adaptation. Inspired by the theoretical analysis of Ben-David et al. [2], learning domain-invariant feature representations has become a

---

[†] Corresponding author.

37th Conference on Neural Information Processing Systems (NeurIPS 2023).

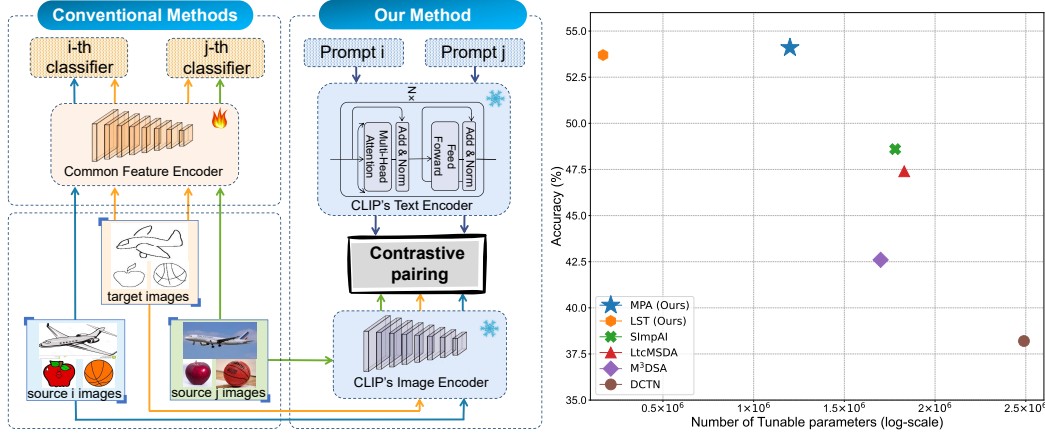

|  |  |
|:---:|:---:|
| (a) Method comparison | (b) Performance comparison on DomainNet |

Figure 1: (a) Most conventional multi-source UDA methods use a common feature extractor with domain-specific classifier heads, while we introduce prompt learning to multi-source UDA. (b) MPA outperforms all other multi-source UDA methods by a large margin on the DomainNet dataset with roughly one-third of tunable parameters. We also introduce an LST strategy for continuous adaptation to a streamlined set of target domains that further reduces the number of tunable parameters and still achieves high accuracy compared with MPA. See texts for more details.

prevailing paradigm for multi-source UDA. One typical approach is to jointly learn a common feature extractor together with domain-specific classifier heads. Various feature distance metrics [22, 33, 14] or domain adversarial training [35] can serve as a preliminary alignment between source and target domains, followed by different auxiliary losses carefully designed to further reduce the domain shift. Although these methods offer decent results in the single source setting, as the number of source domains increases, using only a single shared feature extractor to obtain domain-invariant features is inevitably difficult to optimize [44]. Such a problem is further amplified in the modern large model era if we wish to apply more advanced backbones for improved performance.

In this paper, we introduce prompt learning [16, 41], which has been designed to transfer knowledge learned from large pre-trained vision language models like CLIP [28], to multi-source UDA. In prompt learning, image representations are learned contrastively with a piece of language text termed "prompt". Consequently, prompts are tuned with the backbone network fixed for a more efficient adaptation to downstream tasks (See Figure 1 for a comparison). While recent studies [10, 1] suggest that learnable prompts can be used for UDA, they are restricted to the single source scenario and directly generalizing them to the multi-source setting produces limited results.

In light of this, we present a surprisingly simple framework, **M**ulti-**P**rompt **A**lignment (MPA), for multi-source UDA. MPA is composed of two steps—one to learn an individual prompt by tuning a small set of parameters for each source and target domain pair, while the next to mine the relationships among the learned prompts by deriving a shared embedding space that is domain-invariant. More specifically, given a source domain and a target domain, we use CLIP as our backbone and learn one prompt tailored for such a pair. Then, inspired by the intrinsic dimensionality of data [40], we further reconstruct all the learned prompts using a simple auto-encoder network, aiming at removing redundant information potentially stemmed from the discrepancies among all the source domains. Finally, given the denoised prompts, an $\mathcal{L}_1$ constraint is incorporated as our alignment strategy so that the prompts agree on the classification of target images.

We conduct extensive experiments on multiple benchmarks and the results clearly show that our approach outperforms state-of-the-art methods in the multi-source setting. In particular, on DomainNet [26], the most challenging dataset for multi-source UDA to date, MPA surpasses all state-of-the-art methods. Additionally, as the latent space is optimized with prompts from multiple sources, it encodes knowledge shared by different domains and could potentially generalize to multiple target domains by traversing the space. Consequently, we show how to tune the learned low-dimensional embedding for efficient deployment to a streamlined set of target domains, a strategy we name **L**atent **S**ubspace **T**uning (LST). In summary, our contributions are three-fold:

- We introduce **M**ulti-**P**rompt **A**lignment (MPA) for multi-source UDA. MPA takes advantage of prompt learning and thus is capable of achieving a balance between performance and efficiency compared with alternative methods.
- MPA learns a latent space by maximizing the consensus of multiple learned prompts. Based on this, we introduce **L**atent **S**ubspace **T**uning (LST) that is able to continuously adapt to a streamlined set of target domains.
- MPA achieves state-of-the-art results on several popular benchmarks while LST offers comparable results with tunable parameters that are one order of magnitude less.

## 2 Related Work

### 2.1 Multi-Source Unsupervised Domain Adaptation

First studied by Yang et al. [46], multi-source UDA has drawn increasing attention in the community. Throughout the years, various methods have been studied. For example, in MDAN [48] and DCTN [43], a discriminator applied with adversarial losses is trained so that the features from source and target domains are aligned. MFSAN [51] calculates and aligns the maximum mean discrepancy for each source and target pair. Similarly, M$^3$SDA [26] aligns the moment distance for both target and source domains. All these methods require a shared feature extractor to obtain domain-invariant features. On the contrary, Rakshit et al. [29] adopt one domain-specific encoder for each source and target pair while Zhao et al. [49] pre-trains a classifier for each source domain and then adversarially maps the target images into each trained feature space. The better alignment of these methods is at the cost of a significantly increased number of parameters needed for training. More recently, Xu et al. [44] propose to combine joint alignment and separate alignment for a better adaptation, and Deng et al. [5] propose a method that detects and alleviates the negative impact from pseudo-labels for a better self-training scheme.

### 2.2 Prompt Learning

Traditionally, given a pre-trained language model, a common approach in deep learning is fine-tuning the whole model or its task-specific heads to adjust to downstream tasks. While effective, however, two main drawbacks exist. First of all, as the model size keeps increasing, pre-training and fine-tuning are becoming more and more expensive. Secondly, for each new task, fine-tuning needs to be repeatedly conducted. Recently, researchers have shown that pre-trained large-scale language models can handle a wide range of downstream tasks with only a few or even no samples by pre-pending instructions to the input text [20]. Such instruction texts are called prompts. Consequently, prompts can be tuned instead of the entire network for a more efficient adaptation to downstream tasks. Originally, prompts are essentially sequences of manually designed language tokens that are mapped to an embedding space. To date, extensive research has demonstrated that training soft prompts, *i.e.*, prompts with their own parameters learned by deep models, are more effective [18, 16]. The success of prompt learning in NLP has also garnered attention in the vision community which motivated the establishment of many related works. To name a few, Zhou et al. [50] are the first to apply soft prompt learning to the image recognition task. Ju et al. [13] explore prompt learning for efficient and lightweight video understanding. While promoting in these studies is limited to the input of text encoders, Jia et al. [12] prepend prompt tokens directly to the image patches.

## 3 Method

Our goal is to use multiple labeled source domains for improved performance on target domains. To this end, we leverage prompt learning, which is an effective strategy by learning a small set of parameters to adapt a pretrained model to different downstream tasks. In the following, we first review prompt learning in CLIP in Sec. 3.1 and then elaborate in Sec. 3.2 and Sec. 3.3 our proposed MPA and LST method respectively.

### 3.1 An Overview of Prompt Learning in CLIP

CLIP consists of an image encoder and a text encoder that are jointly trained with a contrastive loss on 400M image and text pairs. The image encoder $f$, which can either be a ResNet [11] or a Vision

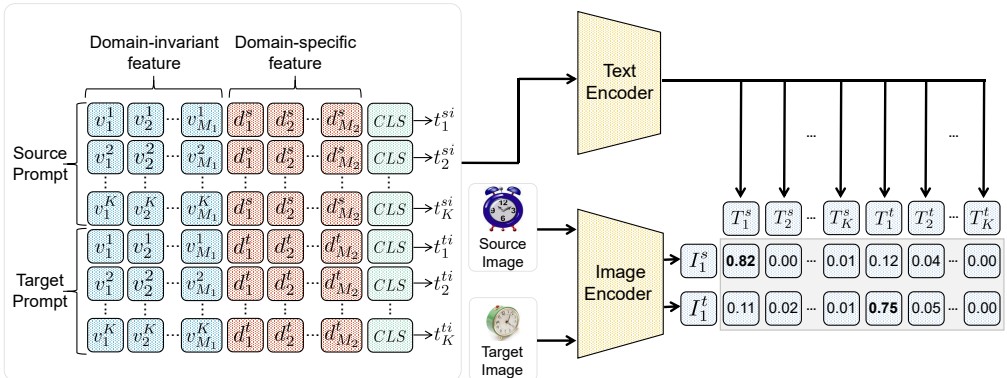

Figure 2: Each source and target pair prompt $\boldsymbol{P}_i$ is the concatenation of a "source prompt" segment and a "target prompt" segment, both composed of domain-invariant and domain-specific features. Therefore, the size of $\boldsymbol{P}_i$ is $\mathbb{R}^{2K \times (M_1 + M_2) \times 512}$. During our prompt training step, the text encoder and the image encoder of CLIP are both frozen.

Transformer [7], maps raw images to an embedding space, and the text encoder $g$ is a Transformer [36] that projects an input text sequence to the same embedding space. A prompt in CLIP usually exists in the form of "a photo of [CLS]" where [CLS] is a class token that can be replaced by a certain class name. This sequence of tokens is first converted into a lower-cased byte pair encoding (BPE) representation, which is essentially a unique numeric ID [50]. Then the numeric IDs are embedded in a 512 dimension vector that is further passed to the Transformer text encoder. In our work, instead of using manually crafted prompts, we train soft prompts that are directly embedded by the text encoder. Given an image $\boldsymbol{x}$ and a text embedding $\boldsymbol{w}_k$ for class $k \in \{1, 2, ..., K\}$, where $K$ is the total number of categories, CLIP aligns them in a contrastive manner so that

$$p(y = k|\boldsymbol{x}) = \frac{\exp(<\boldsymbol{w}_k, f(\boldsymbol{x})>/T)}{\sum_{i=1}^{K} \exp(<\boldsymbol{w}_i, f(\boldsymbol{x})>/T)} \tag{1}$$

is maximized when the input image $\boldsymbol{x}$ indeed belongs to class $k$. Here $< \cdot, \cdot >$ denotes the cosine similarity and $T$ is a learnable temperature parameter.

## 3.2 Multi-Prompt Alignment

Let $N$ denote the total number of domains, where the first $N-1$ domains are source domains and the $N$-th domain is the target domain. For multi-source UDA, our goal is to learn a domain-invariant latent space so that the domain shift among different source domains as well as the discrepancies between all the source and target domain pairs can be minimized. To achieve such a goal, we introduce the framework MPA. Specifically, we design prompts to contain domain-invariant and domain-specific feature following Ge et al. [10] and train such a prompt for each source and target domain pair. Then, an auto-encoder structure is introduced to our method to denoise the acquired prompts, followed by an $\mathcal{L}_1$ constraint for further alignment.

**Prompt Design.** Following Ge et al. [10], our prompt for multi-source UDA includes a set of class-specific context tokens $\boldsymbol{v}_i^k, i \in \{1, 2, ..., M_1\}, k \in \{1, 2, ..., K\}$ and another set of domain-specific tokens shared across all classes $\boldsymbol{d}_j^d, j \in \{1, 2, ..., M_2\}, d \in \{s, t\}$. See Figure 2 for an overview. Here, $M_1$ and $M_2$ represent the number of tokens, $K$ is the number of classes, $s$ is short for source and $t$ is short for target, resulting in a total of $2K$ categories for training with contrastive loss. Therefore, the prompt for each source and target pair can be derived as:

$$\boldsymbol{P}_i = [\boldsymbol{t}_1^{s_i}, ..., \boldsymbol{t}_K^{s_i}, \boldsymbol{t}_1^{t_i}, ..., \boldsymbol{t}_K^{t_i}]^\top, i \in \{1, 2, ..., N-1\}. \tag{2}$$

These prompts serve as learnable parameters that bridge the domain gap between a source domain and a target domain through a contrastive loss, as will be introduced next.

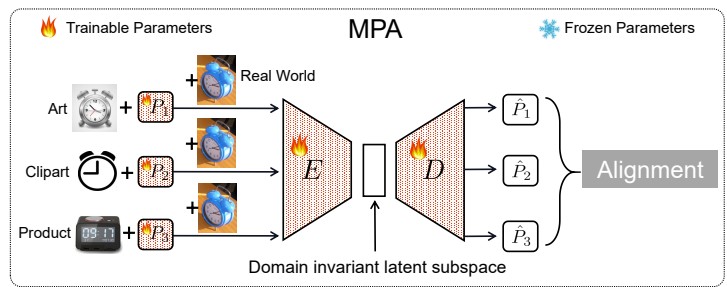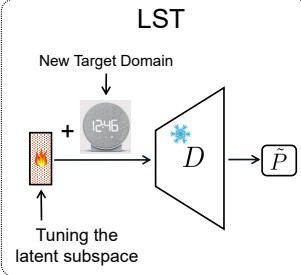

(a) Alignment in MPA on the Office-Home dataset with Rw as target domain  (b) Illustration of LST

Figure 3: (a) Example of prompt alignment on the Office-Home dataset. Here, $\boldsymbol{P}_1, \boldsymbol{P}_2, \boldsymbol{P}_3$ are prompts for domain Ar-Rw, Cl-Rw and Pr-Rw respectively. All prompts are projected into the same latent space for alignment by an auto-encoder structure. (b) When facing a new target domain, tuning the latent subspace learned by the auto-encoder in MPA can allow quick adaptation that is more computationally efficient.

**Prompt Learning for Multi-source UDA.**  To apply prompt learning to multi-source UDA, we first train individual prompts for each source and target pair using the image and text encoders of CLIP. Given an image $\boldsymbol{x}^s$ sampled from the source domain $\mathcal{D}_s$ whose label is $y^*$, we optimize the prompts so that the outputs from the image and text encoder are aligned. For an image $\boldsymbol{x}^t$ from the target domain $\mathcal{D}_t$ whose label is unknown, we first leverage the strong zero-shot ability of CLIP to generate a static pseudo-label $\hat{y^*}$ for image-text alignment. Pseudo-labels are only generated for images whose maximum probability is larger than a fixed threshold $\tau$ using Equation 1. While more sophisticated approaches like self-training could be leveraged to generate pseudo-labels [52, 19], we find that pseudo-labels from CLIP are simple and effective. Finally, prompts are trained with cross-entropy loss functions and Figure 2 gives an overview of the process. More formally, for a prompt $\boldsymbol{P}_i, i \in \{1, 2, ..., N-1\}$, the objective function for optimization follows:

$$\min_{\boldsymbol{P}_i} -\frac{1}{n_s} \sum_{\boldsymbol{x}^s \sim \mathcal{D}_s} \log P(y = y^* | \boldsymbol{x}^s; \boldsymbol{P}_i) - \frac{1}{n_t} \sum_{\boldsymbol{x}^t \sim \mathcal{D}_t} \log P(y = \hat{y^*} | \boldsymbol{x}^t; \boldsymbol{P}_i). \tag{3}$$

Here, the probability $P(y = k | \boldsymbol{x}^d; \boldsymbol{P}_i)$ of an image sample belonging to the $k$-th class is derived from a contrastive loss:

$$P(y = k) = \frac{\exp(< g(\boldsymbol{t}_k^d), f(\boldsymbol{x}^d) > /T)}{\sum_{d \in \{s,t\}} \sum_{i=1}^{K} \exp(< g(\boldsymbol{t}_i^d), f(\boldsymbol{x}^d) > /T)}, \tag{4}$$

where $d \in \{s, t\}$ is a domain identifier indicating where the image comes from, $T$ is a learnable temperature parameter, and $f$ and $g$ represents the image and text encoder in CLIP respectively, which are kept frozen during training. This specific design can push the prompts to learn disentangled representation of both class-invariant and class-specific semantic information to boost the performance of domain adaptation methods [3, 21].

Even though Eqn 3 allows us to obtain a prompt for each source and target domain pair, the noise level in the learned prompts differs due to variations in the amount of images and the domain gap between each source and target domain. One obvious consequence is that they might produce inconsistent results even for the same target domain image. Therefore, an $\mathcal{L}_1$ constraint (Eqn 5) can be applied as a strategy for prompt alignment.

$$\mathcal{L}_1 = \frac{2}{(N-1) \times (N-2)} \sum_{j=1}^{N-2} \sum_{i=j+1}^{N-1} |P(y = k^t | \boldsymbol{x}_k, \boldsymbol{P}_i) - P(y = k^t | \boldsymbol{x}_k, \boldsymbol{P}_j)| \tag{5}$$

**Better Alignment through Reconstructed Prompts.**  While directly aligning the learned prompts produce decent results for multi-source UDA, the prompts are high-dimensional and might contain redundant information. Motivated by the theory that high dimensional data usually lies in a lower

dimensional manifold [40], we build upon auto-encoders for better alignment. By using such an architecture, we hope to learn a domain-invariant "latent subspace" of denoised prompts so that redundant information can be removed through the reconstruction of the learned prompts. More formally, we use an auto-encoder consisting of a projection function $\mathbf{Proj}(\cdot)$ and a back-projection function $\mathbf{Proj}_b(\cdot)$. The learned prompts $\boldsymbol{P}_i$ are first projected into a latent subspace of a lower dimension $d_I$ by $\mathbf{Proj}(\cdot)$, followed by $\mathbf{Proj}_b(\cdot)$ projecting the vectors back into soft prompts $\hat{\boldsymbol{P}}_i$. The $\mathbf{Proj}(\cdot)$ function is implemented by a one-layer feed-forward network while $\mathbf{Proj}_b(\cdot)$ is a two-layer nonlinear perceptron:

$$\mathbf{Proj}(\boldsymbol{P}_i) = \boldsymbol{W}_1(\boldsymbol{P}_i) + \boldsymbol{b}_1 \tag{6}$$

$$\mathbf{Proj}_b(\boldsymbol{v}_I) = \boldsymbol{W}_3(\tanh(\boldsymbol{W}_2\boldsymbol{v}_I + \boldsymbol{b}_1)) + \boldsymbol{b}_2 \tag{7}$$

where $\boldsymbol{v}_I = \mathbf{Proj}(\boldsymbol{P}_i)$ and we optimize a reconstruction loss:

$$\mathcal{L}_{AE} = \frac{1}{N-1}\sum_{i=1}^{N-1}\|\hat{\boldsymbol{P}}_i - \boldsymbol{P}_i\|_2^2. \tag{8}$$

We then perform prompt alignment, *i.e.* the $\mathcal{L}_1$ constraint, on the reconstructed prompts. Therefore, the overall objective function can be written as:

$$\mathcal{L} = \mathcal{L}_{CLS} + \mathcal{L}_{AE} + \alpha\mathcal{L}_1, \tag{9}$$

where $\mathcal{L}_{CLS}$ is the cross entropy loss calculated using the static pseudo-labels that helps with the reconstruction process. Here $\alpha$ is a hyper-parameter controlling the weight of the $\mathcal{L}_1$ loss. The whole alignment procedure is depicted in Figure 3a. Finally, for predicting the labels of target samples, we compute the average of the output logits using all $\hat{\boldsymbol{P}}_i$s.

## 3.3 Latent Subspace Tuning

In real-life applications, it is more practical when adaptation to a streamlined set of targets is needed. While we can repeatedly apply MPA to each target domain, it is computationally inefficient especially when equipped with a large-scale backbone model. To mitigate this issue, we introduce an LST strategy that explores the latent space derived by autoencoders for fast adaptation. The key idea is that since the latent space of the learned auto-encoder from MPA is optimized with prompts from multiple source domains, it alone is capable of encoding domain-invariant knowledge. Therefore, we can adopt MPA on the first target domain, and traverse the subspace learned by MPA to generalize to the following ones.

More formally, given a streamlined set of target domains $\mathcal{D}_{T_1}, \mathcal{D}_{T_2}, ..., \mathcal{D}_{T_L}$, to continuously adapt to each one of them in a computationally efficient manner, we first conduct MPA on domain $\mathcal{D}_{T_1}$. After successfully applying MPA, a low-dimensional embedding space that captures the relationships among different domains is derived and can be leveraged for quick adaptation to domains $\mathcal{D}_{T_2}, ..., \mathcal{D}_{T_L}$. Specifically, for each following target domain $\mathcal{D}_{T_i}$, a domain-invariant feature vector $v_{\text{tune}}^{T_i} \in R^{N \times M_1 \times d_I}$ together with a domain-specific feature vector $d_{\text{tune}}^{T_i} \in R^{1 \times M_2 \times d_I}$ is randomly initialized and passed to the learned back projection function $\mathbf{Proj}_b(\cdot)$ from MPA. As a result, an entirely new prompt $\tilde{\boldsymbol{P}}^{T_i} = \text{Concat}[\mathbf{Proj}_b(v_{\text{tune}}^{T_i}), \mathbf{Proj}_b(d_{\text{tune}}^{T_i})]$ can be constructed. Again with the help of pseudo-labels, we can tune this new prompt by minimizing the following objective function:

$$\min_{d_{\text{tune}}^{T_i}, v_{\text{tune}}^{T_i}} -\frac{1}{n_{T_i}}\sum_{\boldsymbol{x} \sim \mathcal{D}_{T_i}} \log P(y = \hat{y^*}|\boldsymbol{x}; d_{\text{tune}}^{T_i}, v_{\text{tune}}^{T_i}), \tag{10}$$

which is essentially $\mathcal{L}_{CLS}$ in Eqn 9 and inference on $\mathcal{D}_{T_i}$ can be conducted by only using $\tilde{\boldsymbol{P}}^{T_i}$. An illustration of LST is shown in Fig 3b. Since the prompt for CLIP is of size $K \times$ token length $\times$ embedding size, adopting the LST strategy can both reduce the need of training individual prompts and decrease the number of tunable parameters by a factor of at least $(N - 1) \times \frac{\text{embedding size}}{d_I}$ times compared with MPA. This is in fact non-trivial when confronting large-scale datasets such as DomainNet, where $N = 6, d_I = 250$, and embedding size $= 512$.

## 4 Experiments

### 4.1 Experimental Setup

**Datasets and Metrics.** Experiments are conducted on three popular benchmark datasets of UDA to evaluate the effectiveness of MPA, namely ImageCLEF, Office-Home, and DomainNet. ImageCLEF is a small-scaled dataset consisting of 1,800 images from 12 object categories in 3 different domains: ImageNet ILSVRC 2012 (I), Pascal VOC 2012 (P), and Caltech-256 (C). Office-Home is a medium-scaled dataset consisting of about 15,500 images from 65 categories in 4 different domains: Art, Clipart, Product, and Real World. DomainNet is the largest dataset to date, consisting of about 0.6 million images from 345 categories in 6 different domains: Clipart, Infograph, Painting, Quickdraw, Real, and Sketch.

We use top-1 accuracy as our evaluation metric and report results of the following settings: (1) CLIP: zero-shot CLIP on the target domain, which can be regarded as a baseline of our method. (2) Source Combined: all source domains are combined into one single domain and applied with popular single-source UDA methods. Specifically in this setting, we adopt the prompting method from Zhou et al. [50] to serve as another baseline named as "Simple Prompting". (3) Multi-Source: results reported from other multi-source UDA methods.

**Implementation Details** For fair comparisons, we adopt a ResNet50 as our backbone on Image-CLEF and Office-Home and a ResNet101 on DomainNet. The weights are from CLIP and frozen throughout the experiments. Prompts and auto-encoder of MPA are trained using the mini-batch SGD optimizer with a learning rate of 0.003 and 0.005 while the learned subspace is tuned with a 0.0005 learning rate in LST. We use a batch size of 32 and adopt a cosine learning rate scheduler. For hyper-parameters, token lengths $M_1$ and $M_2$ are both set to 16. Pseudo-label threshold $\tau$ is set to 0.4 for producing reliable labels. $\alpha$ in Equation 9 is set to 500. The weight matrix $W_2$ of the back projection function in Equation 7 has a size of $\mathbb{R}^{384 \times d_I}$, where $d_I$ is 100 for ImageCLEF, 150 for OfficeHome and 250 for DomainNet.

### 4.2 Comparison to State-of-the-Art

**Multi-Prompt Alignment** The results on ImageCLEF and Office-Home are shown in Table 1. For ImageCLEF, it is obvious that MPA outperforms other methods with an average accuracy of 91.7%, where there is at least a 3% increase when adapting to domain C and I. For Office-Home, MPA achieves the best results except when adapting to the domain Clipart. Nevertheless, we achieve an accuracy of 75.4% on average, which is 1.3% higher than the second best method MFSAN. It is worth noting that compared to state-of-the-art method MFSAN on both datasets, MPA only trains 0.78M and 2.36M parameters, while MFSAN has a total of 51.75M and 51.80M parameters needed for optimizing (66.3 and 21.9 times larger than ours).

| | ImageCLEF | | | | Office-Home | | | | |
|---|---|---|---|---|---|---|---|---|---|
| | $\rightarrow$ C | $\rightarrow$ I | $\rightarrow$ P | Avg | $\rightarrow$ Ar | $\rightarrow$ Cl | $\rightarrow$ Pr | $\rightarrow$ Rw | Avg |
| **Zero-Shot** | | | | | | | | | |
| CLIP [28] | 95.1 | 87.3 | 74.0 | 85.5 | 71.5 | 50.2 | 81.3 | 82.4 | 71.4 |
| **Source Combined** | | | | | | | | | |
| DAN [22] | 93.3 | 92.2 | 77.6 | 87.7 | 68.5 | 59.4 | 79.0 | 82.5 | 72.4 |
| DANN [9] | 93.7 | 91.8 | 77.9 | 87.8 | 68.4 | 59.1 | 79.5 | 82.7 | 72.4 |
| D-CORAL [33] | 93.6 | 91.7 | 77.1 | 87.5 | 68.1 | 58.6 | 79.5 | 82.7 | 72.2 |
| DAPL* [10] | 96.0 | 89.2 | 76.0 | 87.1 | 72.8 | 51.9 | 82.6 | 83.7 | 72.8 |
| Simple Prompt* | 93.6 | 90.6 | **80.9** | 88.4 | 70.7 | 52.9 | 82.9 | 83.9 | 72.4 |
| **Multi-Source** | | | | | | | | | |
| DCTN [43] | 95.7 | 90.3 | 75.0 | 87.0 | *N.A.* | *N.A.* | *N.A.* | *N.A.* | *N.A.* |
| MDDA [49] | *N.A.* | *N.A.* | *N.A.* | *N.A.* | 66.7 | **62.3** | 79.5 | 79.6 | 71.0 |
| SImpAI$_{50}$ [37] | 93.3 | 91.0 | 77.5 | 87.3 | 70.8 | 56.3 | 80.2 | 81.5 | 72.2 |
| MFSAN [51] | 95.4 | 93.6 | 79.1 | 89.4 | 72.1 | 62.0 | 80.3 | 81.8 | 74.1 |
| **MPA** (ours) | **98.6** | **96.2** | 80.4 | **91.7** | **74.8** | 54.9 | **86.2** | **85.7** | **75.4** |

Table 1: Accuracy (%) on ImageCLEF and Office-Home. * implies that the method is based on our implementation

| | DomainNet | | | | | | |
|---|---|---|---|---|---|---|---|
| | → **Clp** | → **Inf** | → **Pnt** | → **Qdr** | → **Rel** | → **Skt** | **Avg** |
| **Zero-Shot** | | | | | | | |
| CLIP [28] | 61.3 | 42.0 | 56.1 | 10.3 | 79.3 | 54.1 | 50.5 |
| **Source Combined** | | | | | | | |
| DANN [9] | 45.5 | 13.1 | 37.0 | 13.2 | 48.9 | 31.8 | 32.6 |
| MCD [32] | 54.3 | 22.1 | 45.7 | 7.6 | 58.4 | 43.5 | 38.5 |
| DAPL* [10] | 62.4 | 43.8 | 59.3 | 10.6 | 81.5 | 54.6 | 52.0 |
| Simple Prompt* | 63.1 | 41.2 | 57.7 | 10.0 | 75.8 | 55.8 | 50.6 |
| **Multi-Source** | | | | | | | |
| $M^3SDA$-$\beta$ [26] | 58.6 | 26.0 | 52.3 | 6.3 | 62.7 | 49.5 | 42.6 |
| $SImpAI_{101}$ [37] | **66.4** | 26.5 | 56.6 | 18.9 | 68.0 | 55.5 | 48.6 |
| LtC-MSDA [38] | 63.1 | 28.7 | 56.1 | 16.3 | 66.1 | 53.8 | 47.4 |
| T-SVDNet [17] | 66.1 | 25.0 | 54.3 | 16.5 | 65.4 | 54.6 | 47.0 |
| PFSA [8] | 64.5 | 29.2 | 57.6 | **17.2** | 67.2 | 55.1 | 48.5 |
| PTMDA [30] | 66.0 | 28.5 | 58.4 | 13.0 | 63.0 | 54.1 | 47.2 |
| **MPA** (ours) | 65.2 | **47.3** | **62.0** | 10.2 | **82.0** | 57.9 | **54.1** |

Table 2: Accuracy (%) on DomainNet. * implies that the method is based on our implementation

Table 2 shows that for DomainNet, MPA exceeds other multi-source UDA methods by more than 5%. To the best of our knowledge, this is the highest reported accuracy on this dataset so far with less than one third parameters optimized compared with most state-of-the-arts methods. Regarding individual adaptations, MPA achieves best results on most of the adaptation tasks but performs mediocre on the Quickdraw domain. Interestingly, the result is even a little worse than CLIP. We hypothesize that this is because of the large domain gap between Quickdraw and other domains.

**Latent Subspace Tuning**    Results from Table 1 and Table 2 are already strong indications of the success of MPA under conventional multi-source UDA settings. Now, we would like to investigate a more practical scenario, where adaptation to a streamlined set of target domains is required, by using the LST strategy as mentioned in Section 3.3.

Results of LST on the DomainNet dataset are shown in Table 3, where compared with re-applying MPA, performance only dropped by 0.3%. Nevertheless, LST achieves higher accuracy compared to most baseline methods of Table 2, including CLIP with an increase of 3.2%. Significantly, the adaptation process with LST requires only 11 hours of GPU time on a NVIDIA RTX 3090, whereas adapting using MPA takes 54 hours—a speed increase of approximately 5 times.

| | DomainNet | | | | | | |
|---|---|---|---|---|---|---|---|
| | → Inf, **Clp** | → Clp,**Inf** | → Inf, **Pnt** | → Pnt,**Qdr** | → Qdr,**Rel** | → Rel, **Skt** | **Avg** |
| CLIP | 61.3 | 42.0 | 56.1 | 10.3 | 79.3 | 54.1 | 50.5 |
| MPA | 64.9 | 46.6 | 61.8 | 10.2 | 82.5 | 57.5 | 53.9 |
| LST | 64.6 | 46.7 | 61.6 | 9.8 | 81.2 | 57.6 | 53.6 |

Table 3: Results (%) of continuous adaptation on DomainNet. Here → $\mathcal{D}_1, \mathcal{D}_2$ means that adaptation is performed on two target domains and we report the performance on the highlighted second one.

## 4.3 Ablation Study

**CLIP backbone**    While we are the first to apply prompt learning to multi-source UDA with the help of a pre-trained CLIP model, most existing approaches that we compare with utilize ImageNet pre-trained backbones, which are generally considered to be inferior compared to CLIP. Therefore, we would like to examine whether MPA's performance gain is simply from CLIP. For such purpose, two additional sets of experiments are conducted: (1) we apply a simple prompt learning method [50] to the source combined scenario, dubbed as the Simple Prompting baseline. As is demonstrated in Table 1 and  2, while it is 1.33% on average better than zero shot CLIP, MPA still outperforms it with a significant margin; (2) we directly swap MFSAN's ResNet50 backbone pre-trained on ImageNet to CLIP's image encoder. Surprisingly, when tested on ImageCLEF, Table 4 shows that the performance

even drops by a small margin of 0.3%. From these results, we infer that while CLIP indeed provides a strong backbone, it is not necessarily universally superior, as also observed on the basis of studies Devillers et al. [6], Yang et al. [45]. Moreover, despite CLIP's strong zero-shot performance as stated in Radford et al. [28], our method consistently outperforms CLIP in the domain adaptation setting, especially in cases on the ImageCLEF and OfficeHome datasets where CLIP's zero-shot performance is limited. On these two datasets, MPA consistently delivers excellent results that surpass CLIP by 6.2% and 4.0%, respectively. Therefore, we believe while CLIP's strong backbone indeed contributes to MPA's good results, it alone is insufficient for such superior performance and another important piece is MPA's strong domain adaptation ability.

|  | ImageCLEF | | | |
|---|---|---|---|---|
|  | $\rightarrow$ C | $\rightarrow$ I | $\rightarrow$ P | Avg |
| CLIP [28] | 95.1 | 87.3 | 74.0 | 85.5 |
| MFSAN [51] | 95.4 | 93.6 | 79.1 | 89.4 |
| MFSAN+CLIP* | 96.7 | 93.0 | 77.7 | 89.1 |
| **MPA** (ours) | **98.6** | **96.2** | 80.4 | **91.7** |

Table 4: Exploration of MFSAN equipped with CLIP's backbone on ImageCLEF dataset. * implies that the method is based on our implementation

**Prompt engineering**  Apparently, the performance of MPA depends on the quality of pseudo-labels generated by CLIP. As such, it is worth investigating whether MPA has been pushed to its maximum extent by using more sophisticated prompts than "A photo of [CLS]". Specifically, we tested two additional sets of prompts: (1) "a [Dom] of [CLS]" (e.g., a painting of dog) and (2) "a photo of [CLS] in domain [Dom]" (e.g., a photo of dog in domain painting) with the intention of incorporating domain information, and as is presented in Tab 5, while each of the three prompts shows certain advantage on some specific domains, in general, the naive "A photo of [CLS]" still performs the best.

|  | Zero-shot | | | | | MPA | | | | |
|---|---|---|---|---|---|---|---|---|---|---|
|  | $\rightarrow$ Ar | $\rightarrow$ Cl | $\rightarrow$ Pr | $\rightarrow$ Rw | Avg | $\rightarrow$ Ar | $\rightarrow$ Cl | $\rightarrow$ Pr | $\rightarrow$ Rw | Avg |
| a [Dom] of [CLS] | **71.7** | 52.4 | 74.9 | 81.0 | 70.0 | 74.1 | 55.1 | 82.2 | 85.0 | 74.1 |
| a photo of [CLS] in domain [Dom] | 68.1 | **53.1** | 81.3 | 82.0 | 71.1 | 72.7 | **55.7** | **87.1** | 85.2 | 75.2 |
| a photo of [CLS] | 71.5 | 50.2 | 81.3 | **82.4** | **71.4** | **74.8** | 54.9 | 86.2 | **85.7** | **75.4** |

Table 5: Ablation studies on different choices of manual prompt on the Office-Home dataset.

**Hyper-parameter selection**  Experimental results for ablation studies on hyper-parameter selections are reported in Table 6. For both pseudo-label threshold $\tau$ and prompt token length $M_1, M_2$, three different choices $\tau \in \{0.3, 0.6, 0.8\}$ and $M_1, M_2 \in \{8, 12, 20\}$ are examined. For simplicity, we are setting $M_1$ and $M_2$ to be equal. As $\tau$ increases, while the quality of the pseudo-labels gets higher, fewer images will be fed into the model, and Table 6a suggests that doing so hurts the overall performance. On the contrary, shown in Table 6b, the general trend for prompt token length is that the longer the prompt, the better the performance. As a result, we choose $\tau = 0.4$ and $M_1 = M_2 = 16$ to balance the trade-off between performance and efficiency. For $\alpha$ in Equation 9, we chose it to be 500 to balance all losses (in this case $\mathcal{L}_1$) to be of the same order of magnitude. Our experimental results from Table 6c also support such motivation.

**Alignment strategy, prompt denoising and prompt design**  To validate the effectiveness of our alignment strategy, we evaluate our method without the $\mathcal{L}_1$ loss, and table 7 shows that this will result in a performance degradation of about 0.7%. In particular, the $\mathcal{L}_1$ constraint exhibits a greater impact on the more difficult Clipart domain, where the accuracy dropped by a large margin of 1.5% without it, demonstrating the effectiveness of constraining consistent predictions on different prompts. Subsequently, we tested the effectiveness of denoising the prompts through auto-encoder reconstruction. Again, results show that the incorporation of an auto-encoder structure with the $\mathcal{L}_{AE}$ loss is beneficial to the overall performance. Finally, we examined the necessity of using class and domain-specific tokens for learning prompts and results from Table 7 support such design. Specifically, ✓ implies that both the class-specific and domain-specific prompts are used while ✗ implies that only the class-specific prompts are used.

| $\tau$ | $\to$ **Ar** | $\to$ **Cl** | $\to$ **Pr** | $\to$ **Rw** | **Avg** |
|---|---|---|---|---|---|
| 0.3 | 74.5 | **55.0** | 86.1 | **85.9** | **75.4** |
| 0.6 | 74.6 | 54.9 | 85.9 | 85.3 | 75.2 |
| 0.8 | 74.0 | 54.2 | 85.2 | 85.5 | 74.7 |
| 0.4 (reported) | **74.8** | 54.9 | **86.2** | 85.7 | **75.4** |

(a) Ablation on pseudo-label threshold $\tau$.

| Token length | $\to$ **Ar** | $\to$ **Cl** | $\to$ **Pr** | $\to$ **Rw** | **Avg** |
|---|---|---|---|---|---|
| $M_1 = M_2 = 8$ | 74.3 | 54.9 | 85.8 | 85.3 | 75.1 |
| $M_1 = M_2 = 12$ | 74.6 | 54.3 | 85.9 | 85.6 | 75.2 |
| $M_1 = M_2 = 20$ | **74.8** | **55.2** | **86.3** | **86.0** | **75.6** |
| $M_1 = M_2 = 16$ (reported) | **74.8** | 54.9 | 86.2 | 85.7 | 75.4 |

(b) Ablation on token lengths $M_1$ and $M_2$.

| $\alpha$ | $\to$ **Ar** | $\to$ **Cl** | $\to$ **Pr** | $\to$ **Rw** | **Avg** |
|---|---|---|---|---|---|
| 1 | 74.4 | 53.7 | 84.9 | 85.6 | 74.7 |
| 10 | 74.5 | 54.1 | 85.7 | 86.0 | 75.1 |
| 100 | 74.7 | 54.5 | 85.5 | 85.6 | 75.1 |
| 1000 | 74.4 | **55.0** | **86.3** | **86.0** | 75.4 |
| 500 (reported) | **74.8** | 54.9 | 86.2 | 85.7 | **75.4** |

(c) Ablation on $\alpha$.

Table 6: Ablation studies on hyper-parameter selection on the Office-Home dataset.

| $\mathcal{L}_1$ | $\mathcal{L}_{AE}$ | Prompt Design | $\to$ **Ar** | $\to$ **Cl** | $\to$ **Pr** | $\to$ **Rw** | **Avg** |
|---|---|---|---|---|---|---|---|
| ✗ | ✓ | ✓ | 74.8 | 53.4 | 85.2 | 85.3 | 74.7 |
| ✓ | ✗ | ✓ | 74.0 | 53.8 | 85.5 | 85.4 | 74.9 |
| ✓ | ✓ | ✗ | 74.2 | 53.0 | 85.2 | 85.3 | 74.4 |
| ✓ | ✓ | ✓ | 74.8 | **54.9** | **86.2** | **85.7** | **75.4** |

Table 7: Ablation studies on various modules of MPA on the Office-Home dataset.

# 5 Conclusion

In this paper, we introduced prompt learning to multi-source UDA and proposed a simple MPA scheme to align the source and target domains. MPA is composed of two steps. The first step is to train individual prompts for each source and target domain pair and the next one to align them after reconstruction using an auto-encoder structure. Extensive experiments showed that MPA achieved better results than state-of-the-art methods on various multi-source UDA tasks with substantially fewer parameters tuned. Moreover, an LST strategy was introduced for efficient adaptation to a streamlined set of target domains. As the model contains useful clues of multiple domains, one potential limitation is that it faces more risks in terms of information leakage where adversaries might produce attacks to steal information from the model.

**Acknowledgement** This project was supported by NSFC under Grant No. 62102092.

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
