# OpenReview forum: "Multi-Prompt Alignment for Multi-Source Unsupervised Domain Adaptation"
_NeurIPS.cc/2023/Conference — NeurIPS 2023 poster_

### Official Review · Reviewer_34CN · 2023-07-03

**Soundness:** 3 good
**Presentation:** 3 good
**Contribution:** 3 good
**Rating:** 6
**Confidence:** 4

**Summary:**

This paper proposes an interesting pipeline for large pre-trained model-based UDA, and realizes a latent subspace tuning for continuous adaption.


**Strengths:**

1. The motivation behind this article is very meaningful, and the proposed method supports the motivation well.
2. In terms of methodology, the design of this paper is clear and innovative for the combination of existing technologies.
3. This easy but effective baseline and latent subspace tuning ability may provide new paradigm for future UDA.
4. Good performances.

**Weaknesses:**

Using CLIP as the backbone may limit the ability of MPA on only classification tasks, how to extend MPA to other tasks, like segmentation, referring, etc? And how to utilize other large pre-trained models like GPT-3?

**Questions:**

Please see the weakness part.

**Limitations:**

The authors should provide the limitation of their work.

---

> ### Author Rebuttal · Authors · 2023-08-09
>
> > *Using CLIP as the backbone may limit the ability of MPA on only classification tasks, how to extend MPA to other tasks, like segmentation, referring, etc?*
>
> As a matter of fact, there are many other CLIP-based segmentation/referring works [1] [2] [3], all of which used CLIP as backbones. Therefore we believe that CLIP backbone won't be a problem of transfering MPA to other tasks.
>
> [1] Lüddecke1 et al., Image Segmentation Using Text and Image Prompts
>
> [2] Liang et al., Open-Vocabulary Semantic Segmentation with Mask-adapted CLIP.
>
> [3] Wang et al., CRIS: CLIP-Driven Referring Image Segmentation
>
> > *And how to utilize other large pre-trained models like GPT-3?*
>
> One possible way to use other large pre-trained models is to use them for better text prompt generation. However, since CLIP is trained with its own GPT-2 like text encoder, further engineering techniques might be needed for a better adaptation.
>
> > *The authors should provide the limitation of their work.*
>
> Thank you for pointing this out! As a matter of fact, we have provided certain limitation of our work in the Conclusion section: "As the model contains useful clues of multiple domains, one potential limitation is that it faces more risks in terms of information leakage where adversaries might produce attacks to steal information from the model."

---

> ### Author Response · Authors · 2023-08-18
> **Have our rebuttal addressed your concerns?**
>
> Dear reviewer 34CN, we would be grateful if you could confirm whether our response has addressed your concerns. Please do not hesitate to let us know whether there is anything else you would like to see clarified or improved before the end of the rebuttal period.

---

> ### Author Response · Authors · 2023-08-20
> **Have our rebuttal addressed your concerns**
>
> Dear reviewer, as the end of the rebuttal period is approaching, we would like to know whether our rebuttal have addressed your concerns.

---

### Official Review · Reviewer_UUa1 · 2023-07-04

**Soundness:** 4 excellent
**Presentation:** 4 excellent
**Contribution:** 3 good
**Rating:** 5
**Confidence:** 4

**Summary:**

This paper introduces prompt learning to multi-source unsupervised domain adaptation (UDA). Firstly, individual prompts for each source and target pair are learned using a contrastive loss. Then, MPA aligns the learned prompt by an autoencoder-based step with an L_1 constraint to generate consistent results for the same target domain image. In addition, the LST strategy delivers the first target domain adaptation to the subsequent target domains efficiently. Experiments on ImageCLEF, Office-Home, and DomainNet validate the effectiveness.

**Strengths:**

+ This paper's application of prompt learning to multi-source UDA problems is groundbreaking.
+ This paper is well-written and easy to understand.

**Weaknesses:**

+  Regarding the prompt design part in sec3.2, can it only be realized by following Ge[10]? There is a lack of novelty and contribution due to the lack of original works in the whole prompt design.
+  Regarding the problem of reducing the dimensionality of the high-dimensional learned prompt, is this step introduced because the prompt design is not optimal? From the perspective of Tab. 6, the growth brought by AE is very limited, so are other methods of dimensionality reduction effective, or even the step of AE can be removed by considering redundant information in prompt design?
3 The LST in 3 sec 3.3 is a further extended use of MPA, which is somewhat insufficient as an independent innovation point.
4 This paper needs to weaken the sense of the existence of CLIP and the dependence on the previous prompt design method, otherwise very much like a simple application essay.

**Questions:**

In addition to the questions in the above weaknesses that need to be answered, there are a few things that need to be clarified.

+  The generalization of CLIP is a huge advantage. Although the author discussed the impact of CLIP in 4.3, I hope that the author will discuss the design experiment of the prompt. In addition to using the original "a photo of [CLS]," the number of channels of the trainable prompt params can also be discussed.

+ Please state the problems and experimental results faced when the approach of Ge [10] is transferred to the tasks addressed in this paper. And for the problem and results, describe the differences of this paper.

+ For now, it is unknown why prompt, as a trained parameter, has class-specific and domain-specific attributes.

---

> ### Author Rebuttal · Authors · 2023-08-09
>
> > *Regarding the prompt design part in sec3.2, can it only be realized by following Ge[10]?*
>
> There are other ways of implementing the prompt structure. For example, we had considered training just a naive soft prompt for each source-target domain pair, followed by concatenating them and apply an additional linear/convolution layer to extract a "common" prompt for the target domain. However, emperically we find that such way of realizing performs slightly worse than using Ge's:
>
> |             | Art    | Clipart| Product |Real World   | Avg|
> |:-|:-:|:-:|:-:|:-:|:-:|
> |convolutional fusion   |74.1 |54.0|85.3|85.2   |74.7|
> |Ge's |74.8 |54.9 |86.2 |85.7   |75.4|
>
> Yet, we would like to highlight that the success of MPA is not only from such prompt design. As presented in Table 1 and 2, directly extending Ge's work to MDA would result in limited performance.
>
> Furthermore, we had also considered using both text and image prompts. However, since most of the SOTA's are using resnet based backbones, it is hard to incorporate image prompts to such architecture. Therefore for a fair comparison, we chose to not use such strategy.
>
> > *Regarding the problem of reducing the dimensionality of the high-dimensional learned prompt, is this step introduced because the prompt design is not optimal? From the perspective of Tab. 6, the growth brought by AE is very limited, so are other methods of dimensionality reduction effective, or even the step of AE can be removed by considering redundant information in prompt design?*
>
> Thank you for the question! We would like to note that our intention is **reconstruction** rather than **dimension reduction**, and the reason is not because of the prompt design, but to "remove redundant information potentially stemmed from the discrepancies among all the source domains.", as stated in L55-57. This is one major reason why we kept the AE structure. Another reason is that by leveraging the auto-encoder structure, especially its decoder, we are able to adopt our LST strategy that allows efficient and effective adaptation to multiple target domains, which we believe is of practical importance.
>
> > *The LST in 3 sec 3.3 is a further extended use of MPA, which is somewhat insufficient as an independent innovation point.*
>
> We respectfully disagree that LST doesn't serve as an innovation point. While LST is based on the domain-invariant latent space found by MPA, it surves different purpose and has a complete different design principle compared with MPA. Specifically, LST is most suitable in situation where adaptation to multiple target domains is needed. In such scenarios, MPA would require **all** source-target domain prompts to be repeatedly trained. On the contrary, for LST, all we need is to tune **one** prompt with only the target domain data, which significantly reduces the computational cost. As stated in L249, LST would boost the speed of adaptation on the DomainNet dataset of approximately 5 times. Such improvement is far beyond marginal and we humbly believe that it serves as a solid contribution.
>
> > *This paper needs to weaken the sense of the existence of CLIP and the dependence on the previous prompt design method, otherwise very much like a simple application essay.*
>
> Thank you for the suggestion! We will revise accordingly.
>
> > *I hope that the author will discuss the design experiment of the prompt. In addition to using the original "a photo of [CLS]," the number of channels of the trainable prompt params can also be discussed.*
>
> As a matter of fact, the channels of prompt parameters are fixed to 512-d in CLIP, as stated in L122. What could be changed is the length of the prompt token, i.e., the $M_1$ and $M_2$ in L140, and ablation study on these parameters are discussed in Section 4.3 (Table 5).
>
> > *Please state the problems and experimental results faced when the approach of Ge [10] is transferred to the tasks addressed in this paper. And for the problem and results, describe the differences of this paper.*
>
> The usual way of extending single source method to multiple sources (the task in this paper) is to apply to the source combined scenario, where all source domains are combined into one single domain, and this is what we have done in the paper (DAPL in Source Combined). However, by doing so, one major problem is that no strategy is taken on dealing with domain gap among the source domains, and often the times will produce unsatisfactory performances. The experimental results for doing so are already shown in Table 1 and 2, and we have put them on the tables below for reference:
>
> ImageCLEF:
> |           | C   | I   | P   | Avg   |
> |:-|:-:|:-:|:-:|:-:|
> |DAPL in Source Combined      | 96.0|89.2 |76.0.7 | 87.1  |
> |MPA     | 98.6|96.2 |80.4 |  91.7 |
>
> OfficeHome:
> |             | Art   | Clipart   | Product   | Real World   | Avg |
> |:-|:-:|:-:|:-:|:-:|:-: |
> |DAPL in Source Combined   |72.8 |51.9 |82.6 |83.7   | 72.8 |
> |MPA|74.8 |54.9 |86.2 |85.7   | 75.4|
>
> DomainNet:
> |           |  Clp |   Inf   | Pnt    |  Qdr   | Rel  | Skt | Avg|
> |:-|:-:|:-:|:-:|:-:|:-:|:-:|:-:|
> |DAPL in Source Combined         |62.4 |43.8 |59.3 |10.6   |81.5|54.6 | 52.0 |
> |MPA         |65.2 |47.3 |62.0 |10.2   |82.0 |57.9 | 54.1 |
>
> Here we can see MPA surpasses DAPL in Source Combined setting by an average of 3.1%, indicating the efficacy of our approach. The reason for such performance gain is that unlike DAPL, in MPA we treated each source domain and target domain pair independently, and further incorporated our alignment strategy for dealing with the domain gap among the source domains.
>
> > *For now, it is unknown why prompt, as a trained parameter, has class-specific and domain-specific attributes.*
>
> Great question! The intuition is that if we use a hard prompt like "a [Domain] of [CLS]", then for each class, the embedding of [Domain] will remain the same while that of [CLS] will change. We hope this is what the soft prompt will learn through training.

---

> > ### Comment · Reviewer_UUa1 · 2023-08-20
> > **Additional Comments**
> >
> > Thank you to the author for the careful responses that solved most of my questions. I'm keeping my score unchanged, mainly considering the lack of novelty of the prompt method proposed and the closeness of the implementation of LST to MPA. I  appreciate the motivation for this article, which is the main reason for me to hold the current opinion.  It would be better to do some prompt and structural innovation.

---

> ### Author Response · Authors · 2023-08-18
> **Have our rebuttal addressed your concerns?**
>
> Dear reviewer UUa1, we would be grateful if you could confirm whether our response has addressed your concerns. Please do not hesitate to let us know whether there is anything else you would like to see clarified or improved before the end of the rebuttal period.

---

> ### Author Response · Authors · 2023-08-20
> **Have our rebuttal addressed your concerns?**
>
> Dear reviewer, as the end of the rebuttal period is approaching, we would like to know whether our rebuttal have addressed your concerns.

---

### Official Review · Reviewer_tF9m · 2023-07-05

**Soundness:** 2 fair
**Presentation:** 2 fair
**Contribution:** 2 fair
**Rating:** 5
**Confidence:** 4

**Summary:**

This paper deals with the multi-source domain adaptation problem. It proposes to tune the designed domain-invariant prompts and domain-specific prompts to enable the domain adaptation ability. Generally, the training consists of two objectives, i.e., the individual prompt learning objective and the de-noising objective via prompt auto-encoders. To enable a test-time domain adaptation ability and reduce the amount of learnable parameters, the authors propose LST. Experiments on various multi-source benchmarks verify the effectiveness of proposed method.

**Strengths:**

- The idea is simple and generally reasonable.

- The experiment results are good.

**Weaknesses:**

- In LST, I don't think it is reasonable to use randomly initialized representations as the input of the back-projector to perform prompt reconstruction, as the latent representations don't subject to the same distribution. And there is no empirical evidence to show such a way really works.

- Missing comparisons to strong baselines. E.g., on DomainNet, [1] achieves a multi-source domain adaptation performance at 53.2 with a ResNet-101 backbone. The authors should compare to the strong baselines.

  [1] Contrastive adaptation network for single- and multi-source domain adaptation, TPAMI 2020.

**Questions:**

See the weakness part.

**Limitations:**

I don't see any serious potential negative societal impact of this work.

---

> ### Author Rebuttal · Authors · 2023-08-09
>
> > *In LST, I don't think it is reasonable to use randomly initialized representations as the input of the back-projector to perform prompt reconstruction, as the latent representations don't subject to the same distribution. And there is no empirical evidence to show such a way really works.*
>
> As pointed out in L130, the learned latent space is supposed to be domain-invariant. Besides, this representation is further tuned by pseudolabels of the target data. Therefore even though the representations subject to different distribution, the final result shouldn't be affected.
>
> As for empirical evidence, there are actually a few research papers that would use a random initialization when tuning the latent space. For example, in [1], 1000 latent vectors are randomly initialized to find the best performing one. In [2], it is found that random initialized representations are better at generating pictures other than faces. Here the representations are also of a different distribution (faces v.s. cat, dog etc.).
>
> [1] Wen et al., Diamond in the rough: Improving image realism by traversing the GAN latent space
>
> [2] Abdal et al., Image2StyleGAN: How to Embed Images Into the StyleGAN Latent Space?
>
> > *Missing comparisons to strong baselines. E.g., on DomainNet, [1] achieves a multi-source domain adaptation performance at 53.2 with a ResNet-101 backbone. The authors should compare to the strong baselines.*
>
> Thank you for pointing this out! We will include [1] in our paper for a more comprehensive comparison. We would also like to note that since [1] performs a clustering on all source domains for each iteration, our method is much more efficient.
>
> |           |  Clp |   Inf   | Pnt    |  Qdr   | Rel  | Skt | Avg|
> |:------------|:-----:|:---:|:---:|:-----:|---:|---:|---:|
> |      MSCAN   |69.3 |28.0 |58.6 |  30.3 |73.3|59.5 | 53.2 |
> |MPA         |65.2 |47.3 |62.0 |10.2   |82.0 |57.9 | 54.1 |
>
> [1] Kang et al., Contrastive adaptation network for single- and multi-source domain adaptation.

---

> ### Author Response · Authors · 2023-08-18
> **Have our rebuttal addressed your concerns?**
>
> Dear reviewer tF9m, we would be grateful if you could confirm whether our response has addressed your concerns. Please do not hesitate to let us know whether there is anything else you would like to see clarified or improved before the end of the rebuttal period.

---

> ### Author Response · Authors · 2023-08-20
> **Have our rebuttal addressed your concerns?**
>
> Dear reviewer, as the end of the rebuttal period is approaching, we would like to know whether our rebuttal have addressed your concerns.

---

### Official Review · Reviewer_tSjx · 2023-07-09

**Soundness:** 3 good
**Presentation:** 3 good
**Contribution:** 3 good
**Rating:** 6
**Confidence:** 4

**Summary:**

The paper proposes an extension of [10] (Domain Adaptation via Prompt Learning Ge et al., 2022) to the multi-source UDA set-up.
(i) Distinct soft prompts are learnt via contrastive loss for each source-target pair;  each source-target prompt is composed of class-wise source- and target-prompts. In the target domain, learning is achieved by utilizing pseudo-labels from the pretrained CLIP model, which possesses strong zero-shot ability.
To encourage consistent target outputs across all learned prompts, a consistency L1 loss is applied to the soft outputs of all target samples.
(ii) Prompt reconstruction: to eliminate redundant information that may impede performance, an autoencoder (AE) is learned to "denoise" the acquired prompts.
(iii) Latent Subspace Tuning (LST): efficient adaptation to a new target domain, after learning from the first target domain, is accomplished by optimizing on the latent space of the learned AE using pseudo-labels of the new target.

The proposed framework demonstrates superior results compared to previous State-of-the-Art (SOTA) methods in multi-source UDA.

**Strengths:**

The proposed method leverages the powerful zero-shot capability of the pretrained CLIP model for the multi-source domain adaptation (DA) task. Empirical results demonstrate its effectiveness across various benchmarking setups. The presentation is good, enabling easy comprehension of the method.

**Weaknesses:**

The main concern of this work is the lack of technical novelty and a more rigorous evaluation. The proposed framework is a straightforward extension of the work by Get et al. (2022) [10] to the multi-source setting. In terms of method evaluation, it is expected to compare against stronger CLIP-based baselines, both in the multi-source UDA experiments (Tables 1 and 2) and in the LST experiment (Table 3). Further details are provided in the following section.

**Questions:**

Given that CLIP alone produces results close to the state-of-the-art (SOTA) as shown in Table 1, and even outperforms the SOTA as shown in Table 2, my main concern is whether the authors have made adequate efforts to create decent baselines using CLIP-pretrained models. I appreciate their effort in Section 4.3 when they ``swap MFSAN’s ResNet50 backbone pre-trained on ImageNet to CLIP’s image encoder''. However, I would love to know more details to know as if careful considerations were taken into account for MFSAN + CLIP or simply the authors swap the backbone's weight. For example, did the authors preserve the text classifier and the contrastive los of CLIP ? As the proposed framework is based on self-training with pseudo-labels, I wonder if how the CLIP + self-training and Single Prompt + self-training baselines perform.

Furthermore, I am curious to know if CLIP with fixed prompts has been pushed to its maximum extent. In other words, how far can this naïve baseline reach with better prompt engineering instead of using the simple prompt "a photo of [CLS]"? One can imagine incorporating domain-specific information into the prompt or defining a set of templates rather than relying on just one template.

In the LST experiment, how good is MPA (on the first domain) + self-training? For efficiency, one can try test time prompt tuning (TPT).

**Limitations:**

This work lacks significant technical novelty, and the experiment could be improved by using stronger baselines. Based on the current state of the submission, my recommendation leans towards the negative side.

---- After rebuttal ----
The rebuttal is convincing and has helped clarify most of my technical concerns. I believe that this work is indeed interesting and help advocate the usage of prompts in domain adaptation. While the novelty limitation still pertains, the original set of experiments, along with the new ones provided during the rebuttal, is sufficient. I believe the paper after revision would certainly pass the NeurIPS' bar.

---

> ### Author Rebuttal · Authors · 2023-08-09
>
> > *Lack of technical novelty. The proposed framework is a straightforward extension of the work by Ge et al. to the multi-source setting.*
>
> We respectfully disagree. As shown in Table 1 and 2, directly applying Ge's method to MDA produces limited performance. This is because their method lacks strategies on dealing with domain gap among the source domains. On the contrary, Table 6 shows that with our proposed alignment strategy, domain shift among source domains is effectively reduced, resulting in a significant performance boost.
>
> > *Lack of a more rigorous evaluation. It is expected to compare against stronger CLIP-based baselines.*
>
> To the best of our knowledge, there are **no** other CLIP-based MDA baselines. To make up for such shortcomings, we proposed **3** CLIP-based baselines, i.e., the zero-shot CLIP baseline, the DAPL in the source combined scenario baseline, and a Simple Prompt baseline based on the paper of CoOp. Results from Table 1 and 2 show that MPA consistently achieves better results than these methods.
>
> > *Given that CLIP alone produces good results, have the authors made adequate efforts to create decent baselines using CLIP-pretrained models.*
>
> We would like to highlight again that there are no other CLIP-pretrained baselines. Therefore, we believe **CLIP itself forms a decent baseline**. In addition to CLIP, we have also proposed a Simple Prompt baseline, and tested a SOTA method with CLIP-pretrained backbone. More details on these baselines are in the following answers.
>
> We understand that concerns might be raised by using CLIP and really have tried our best to present a fair comparison. If you feel that these are insufficient, would you please suggest us what other attempts we can do?
>
> > *Whether careful considerations were taken into account for MFSAN + CLIP?*
>
> We've actually tried keeping the text classifier with the contrastive loss, but finally decided to not do so. The reasons are as follows.
>
> In MFSAN, a common feature extractor together with domain specific feature extractors and domain specific classifier heads are used. In order to keep the text classifier, two main technical difficulties need to be overcomed: (1) Text classifier generated by CLIP is dependent on the input text prompt and would be the same for all domains if we use the naive "a photo of a [CLS]" prompt, which contradicts the design of MFSAN where domain specific classifier heads are used; (2) In MFSAN, the features extracted from the common feature extractor are downsampled from 2048-d to 256-d, whereas text classifier from CLIP is 1024-d.
>
> To solve the above issues, we could apply soft prompt methods and train a prompt for each domain to generate the domain specific heads, followed by either applying another linear layer to reduce their dimension to 252-d or only downsample the features to 1024-d. However, both settings produced unsatisfactory results:
>
> |           | C   | I   | P   | Avg   |
> |:-|:-:|:-:|:-:|:-:|
> |Linear layer to 252-d      | 21.0|37.8 |19.7 | 26.2  |
> |Downsample to 1024-d     | 88.7|81.7 |74.2 |  81.5 |
> |No text classifier |96.7 |93.0 |77.7 |89.1   |
>
> Based on the above results, further techniques might be needed to acquire a good performance, which however, would most likely result in a complete new method and is beyond the scope of analyzing whether the performance gain of MPA is from CLIP's pretrained visual backbone. Therefore, we chose not to keep the text classifier and only replaced the weights of the common feature extractor with CLIP's visual backbone weights.
>
> > *How the CLIP + self-training and Single Prompt + self-training baselines perform.*
>
> When using naive CLIP with manually designed prompts, there are no parameters needed for training (usually the image backbone is frozen). Therefore it is rather unclear to us what CLIP + self-training refers to. Did you mean CLIP with soft prompts + self-training? If so, this is exactly what the Simple Prompt baseline refers to in Table 1 and 2. In Simple Prompt we train a soft prompt using pseudolabels generated by CLIP and results show that simply doing so is ineffective in achieving good MDA ability.
>
> We are also confused about the "Single Prompt + self-training baseline". Did you mean "Simple Prompt + self-training"? If so, in Simple Prompt, self-training is already used.
>
> > *If CLIP with fixed prompts has been pushed to its maximum extent.*
>
> Thank you for the suggestion! Based on your comment, we tested two sets of prompts: (1) "a [Domain] of [CLS]" (e.g., a painting of dog) and (2) "a photo of [CLS] in domain [Domain]" (e.g., a photo of dog in domain painting). Their zero-shot results on Office-Home are shown in the following table:
>
> |             | Art   | Clipart   | Product   | Real World   | Avg |
> |:-|:-:|:-:|:-:|:-:|:-: |
> |a [Domain] of [CLS]         |71.7 |52.4 |74.9 |81.0   | 70 |
> |a photo of [CLS] in domain [Domain]|68.1 |53.1 |81.3 |82.0   | 71.1|
> |a photo of [CLS]    |71.5 |50.2 |81.3 |82.4   |71.4|
>
> We also tested MPA's performance with pseudolabels generated from these two new prompts:
>
> |             | Art   | Clipart   | Product   | Real World   | Avg |
> |:-|:-:|:-:|:-:|:-:|:- |
> |a [Domain] of [CLS]         |74.1 |55.1 |82.2 |85.0   |74.1|
> |a photo of [CLS] in domain [Domain]|72.7 |55.7 |87.1 |85.2   |75.2|
> |a photo of [CLS]    |74.8 |54.9 |86.2 |85.7   |75.4|
>
> In general, the default "a photo of [CLS]" still performs the best. While better handcrafted prompts might exist, for now due to time limitations we are unable to find significantly superior ones.
>
> > *In the LST experiment, how good is MPA (on the first domain) + self-training?*
>
> MPA on the first domain already uses self-training. The results are slightly worse than those in Table 2, as less source domains are used:
>
> |           | &rarr; Inf*,Clp   | &rarr; Clp*,Inf    | &rarr; Skt*, Pnt     | &rarr; Pnt*, Qdr     | &rarr; Qdr*, Rel   | &rarr; Rel*, Skt  |
> |:-|:-:|:-:|:-:|:-:|-:|:-:|
> |MPA         |46.7 |64.6 |57.5 |62.2   |10.1|81.8 |

---

> ### Author Response · Authors · 2023-08-18
> **Have our rebuttal addressed your concerns?**
>
> Dear reviewer tSjx, we would be grateful if you could confirm whether our response has addressed your concerns. Please do not hesitate to let us know whether there is anything else you would like to see clarified or improved before the end of the rebuttal period.

---

> ### Author Response · Authors · 2023-08-20
> **Have our rebuttal addressed your concerns?**
>
> Dear reviewer, as the end of the rebuttal period is approaching, we would like to know whether our rebuttal have addressed your concerns.

---

### Decision · Program_Chairs · 2023-09-21

**Decision:**

Accept (poster)

**Comment:**

The reviewers all agreed that the problem is important and that the method is sound. Several reviewers expressed reservations about the novelty of the method, namely the differences to [10]. They also raised several questions of detail. The authors provided a rebuttal that addressed most of the reviewer concerns. After discussion, all reviewers were positive towards publication, although none was strongly supportive. It was nevertheless considered that the paper will make an interesting contribution.